# SHALLOW LEARNING FOR DEEP NETWORKS

## ABSTRACT

Shallow supervised 1-hidden layer neural networks have a number of favorable properties that make them easier to interpret, analyze, and optimize than their deep counterparts, but lack their representational power. Here we use 1-hidden layer learning problems to sequentially build deep networks layer by layer, which can inherit properties from shallow networks. Contrary to previous approaches using shallow networks, we focus on problems where deep learning is reported as critical for success. We thus study CNNs on image recognition tasks using the large-scale ImageNet dataset and the CIFAR-10 dataset. Using a simple set of ideas for architecture and training we find that solving sequential 1-hidden-layer auxiliary problems leads to a CNN that exceeds AlexNet performance on ImageNet. Extending our training methodology to construct individual layers by solving 2-and-3-hidden layer auxiliary problems, we obtain an 11-layer network that exceeds VGG-11 on ImageNet obtaining $89.8\%$ top-5 single crop. To our knowledge, this is the first competitive alternative to end-to-end training of CNNs that can scale to ImageNet. We conduct a wide range of experiments to study the properties this induces on the intermediate layers.

## 1 INTRODUCTION

Deep Convolutional Neural Networks (CNNs) trained on large-scale supervised data via the back-propagation algorithm have become the dominant approach in most computer vision tasks (Krizhevsky et al., 2012). This has motivated successful applications of deep learning in other fields such as speech recognition (Chan et al., 2016), natural language processing (Vaswani et al., 2017), and reinforcement learning (Silver et al., 2017). Training procedures and architecture choices for deep CNNs have become more and more entrenched, but which of the standard components of modern pipelines are essential to the success of deep CNNs is not clear. Here we pose the question: do CNN layers need to be learned end-to-end to obtain high performance? We will show even for the complex imagenet dataset the answer is possibly no.

Supervised end-to-end learning is the standard approach to neural network optimization. However it has potential issues that can be valuable to consider. First, the use of a global objective means that the final functional behavior of individual intermediate layers of a deep network is only indirectly specified: it is entirely unclear how the layers work together to achieve high-accuracy predictions. Several authors have suggested and shown empirically that CNNs learn to implement mechanisms that progressively induce invariance to complex, but irrelevant variability (Mallat, 2016; Yosinski et al., 2015) while increasing linear separability (Zeiler & Fergus, 2014; Oyallon, 2017; Jacobsen et al., 2018) of the data. Progressive linear separability has been shown empirically but it is unclear whether this is merely the consequence of other strategies implemented by CNNs, or if it is a sufficient condition for the observed high performance of these networks. Secondly, understanding the link between shallow Neural Networks (NNs) and deep NNs is difficult: while generalization, approximation, or optimization results (Barron, 1994; Bach, 2014; Venturi et al., 2018; Neyshabur et al., 2018; Pinkus, 1999) for 1-hidden layer NNs are available, the same studies conclude that multiple-hidden-layer NNs are much more difficult to tackle theoretically. Finally, end-to-end back-propagation can be inefficient (Jaderberg et al., 2016; Salimans et al., 2017) in terms of computation and memory resources. Moreover, for some learning problems, the full gradient is less informative than other alternatives (Shalev-Shwartz et al., 2017).

Sequential learning of CNN layers by solving shallow supervised learning problems is an alternative to end-to-end back-propagation. This strategy can directly specify the objective of every layer for

example by encouraging the refinement of specific properties of the representation (Greff et al., 2016), such as progressive linear separability. The development of theoretical tools for deep greedy methods could then draw from the theoretical understanding of shallow sub-problems. Indeed, Arora et al. (2018); Bengio et al. (2006); Bach (2014); Janzamin et al. (2015) show global optimal approximations, while other works have shown that networks based on 1-hidden layer training can have a variety of guarantees under certain assumptions (Huang et al., 2017; Malach & Shalev-Shwartz, 2018; Arora et al., 2014): greedy layerwise methods could permit to cascade those results to bigger architectures. Finally, a greedy approach will rely much less on having access to a full gradient. This can potentially avoid pathologies such as in Shalev-Shwartz et al. (2017). From an algorithmic perspective, they do not require storing most of the intermediate activations nor to compute most intermediate gradients. This can be beneficial in memory-constrained settings. Unfortunately, prior work has not convincingly demonstrated that layerwise strategies can tackle the sort of large scale problems that have brought deep learning into the spotlight. We propose a straightforward strategy for CNNs that is shown to scale and analyze the representations it builds.

Our contributions are as follows. **(a)** First, we design a simple and scalable supervised approach to learn layer-wise CNNs in Sec. 3. **(b)** Then, Sec. 4.1 demonstrates empirically that by sequentially solving 1-hidden layer problems, we can match the performance of the AlexNet on ImageNet. This supports a body of literature that tackle 1-hidden layer networks and their sequentially trained counterparts. **(c)** We show that layerwise trained layers exhibit a progressive linear separability property in Sec. 4.2. **(d)** In particular, we use this to help motivate learning layer-wise CNN layers via shallow $k$-hidden layer auxiliary problems, with $k > 1$. Using this approach our sequentially trained 3-hidden layer models can reach the performance level of VGG-13 (Sec. 4.3). **(e)** Finally, we suggest an approach to easily reduce the model size *during training* of these networks.

## 2 RELATED WORK

Several authors have previously considered layerwise learning. In this section we review several of the related works and re-emphasize the distinctions from our work.

Greedy unsupervised learning has been a popular topic of research in the past. Greedy unsupervised learning of deep generative models (Bengio et al., 2007; Hinton et al., 2006) was shown to be effective as an initialization for deep supervised architectures. Bengio et al. (2007) also considered supervised greedy layerwise learning as *initialization* of networks for subsequent end-to-end supervised learning, but this was not shown to be effective with the existing techniques at the time. Later work on large-scale supervised deep learning showed that modern training techniques permit avoiding layerwise initialization entirely (Krizhevsky et al., 2012). We emphasize that the supervised layerwise learning we consider is distinct from unsupervised layerwise learning. Moreover, here layerwise training is not studied as a *pretraining* strategy, but a *training* one.

Layerwise learning in the context of constructing supervised NNs has been attempted in several works. Early demonstrations have been made in Fahlman & Lebiere (1990b); Lengellé & Denoeux (1996) on very simple problems and in a climate where deep learning was not a dominant supervised learning approach. These works were aimed primarily at structure learning, building up architectures that allow the model to grow appropriately based on the data. Similarly, Cortes et al. (2016) recently proposed a progressive learning method that builds a network such that the architecture can adapt to the problem, with focus on the theory associated with the structure learning problem, but do not consider problems where deep networks are unmatched. Malach & Shalev-Shwartz (2018) also train a supervised network in a layerwise fashion, showing that their method provably generalizes for a restricted class of image models. However, the results of these model are not shown to be competitive with handcrafted approaches (Oyallon & Mallat, 2015). Similarly (Kulkarni & Karande, 2017) consider a kernel based layerwise objective, but with unconvincing results in very limited settings.

Boosting techniques (Friedman, 2001; Freund et al., 1996) are a greedy approach to supervised learning with a successful history and theoretical foundation and still represents the state of the art in some domains (Chen & Guestrin, 2016). Recently Huang et al. (2017) combined boosting theory with a modern residual network (He et al., 2016) by sequentially training layers. The properties of the residual are exploited to effectively leverage boosting theory. However, results are presented for limited datasets and indicate that the end-to-end approach is often needed ultimately to obtain

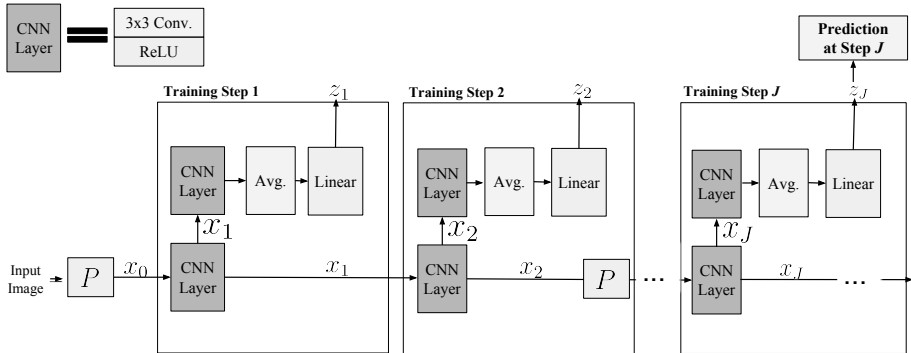

Figure 1: High level diagram of our layer-wise learning framework using a $k = 2$-hidden layer. $P$, the down-sampling (Jacobsen et al., 2018, Fig. 2), is applied at the input image as well as at $j = 2$.

competitive results. The proposed sequential strategy does not clearly outperform simple non-deep learning baselines. By contrast our work focuses on settings where CNN based-approaches do not currently have competitors and introduces the use of auxiliary hidden layers.

Another related thread are methods which add layers to existing networks and then use end-to-end learning. These approaches usually have different goals from ours, such as stabilizing end-to-end learned models. Brock et al. (2017) builds a network in stages, where certain layers are progressively frozen, which permits faster training. Mosca & Magoulas (2017); Wang et al. (2017) propose methods that stack progressively layers at each turn performing, end-to-end learning on the resulting network. A similar strategy was applied for training GANs in Karras et al. (2017). By the nature of our objectives in this work, we never perform fine-tuning of the whole network. Finally several methods consider auxiliary supervised objectives Lee et al. (2015) to stabilize end-to-end learning, but which is rather different from the case where these objectives are not solved jointly.

## 3 SUPERVISED LAYERWISE TRAINING OF CNNS

In this section we formalize the architecture, training algorithm, and the necessary notations and terminology. We focus on CNNs, with ReLU non-linearity denoted by $\rho$. Sec. 3.1 will describe a layer-wise training scheme using a succession of auxiliary learning tasks. We add one layer at a time: the first layer of a $k$-hidden layer CNN problem. Finally, we will discuss the distinctions in varying $k$.

### 3.1 ARCHITECTURE FORMULATION

Our architecture has $J$ blocks (see Fig. 1), which are trained in succession. From an input signal $x$, an initial representation $x_0 \triangleq x$ is propagated through $j$ convolutions, giving $x_j$. Each $x_j$ feeds into an *auxiliary classifier* to obtain prediction $z_j$, which computes an intermediate classification output. At depth $j$, denote by $W_{\theta_j}$ a convolutional operator with parameters $\theta_j$, $C_{\gamma_j}$ an auxiliary classifier with all its parameters denoted $\gamma_j$, and $P_j$ a down-sampling operator. The parameters correspond to $3 \times 3$ kernels with bias terms. Formally, from layer $x_j$ we construct $\{x_{j+1}, z_{j+1}\}$ as follows:

$$\begin{cases} x_{j+1} = \rho W_{\theta_j} P_j x_j \\ z_{j+1} = C_{\gamma_j} x_{j+1} \in \mathbb{R}^c \end{cases} \tag{1}$$

where $c$ is the number of classes. For the pooling operator $P$ we choose the *invertible downsampling* operation described in Dinh et al. (2017), which consists in reorganizing the initial spatial channels into the 4 spatially decimated copies obtainable by $2 \times 2$ spatial sub-sampling, reducing the resolution by a factor 2. We decided against strided pooling, average pooling, and the non-linear max-pooling, because these strongly encourage a loss of information. As is standard practice in CNNs, $P$ is applied at certain layers ($P_j = P$), but not others ($P_j = Id$). The classifier $C_{\gamma_j}$ is a CNN that can be written:

$$C_{\gamma_j} x_j = \begin{cases} LA x_j & \text{for } k = 1 \\ LA\rho \tilde{W}_{k-2}...\rho\tilde{W}_0 x_j & \text{for } k > 1 \end{cases} \tag{2}$$

where $\tilde{W}_0, ..., \tilde{W}_{k-2}$ are convolutional layers with constant width, $A$ is a spatial averaging operator, and $L$ a linear operator whose output dimension is $c$. We remark the averaging operation is important for maintaining scalability at early layers. Observe that for $k = 1$, $C_{\gamma_j}$ is simply a linear model, and in this case our architecture will be trained by a sequence of 1-hidden layer CNN.

## 3.2 Training by Auxiliary Problems

Our training procedure is layerwise: at depth $j$, while keeping all other parameters fixed, $\theta_j$ is obtained via an *auxiliary problem*: optimizing $\{\theta_j, \gamma_j\}$ to obtain the best training accuracy for *auxiliary classifier* $C_{\gamma_j}$. We now formalize this idea for a training set $\{x^n, y^n\}_{n \leq N}$. For a function $z(\cdot; \theta, \gamma)$ parametrized by $\{\theta, \gamma\}$ and a loss $l$ (e.g. cross entropy), we consider the classical minimization of the empirical risk:

$$\hat{\mathcal{R}}(z; \theta, \gamma) \triangleq \frac{1}{N} \sum_n l(z(x^n; \theta, \gamma), y^n)$$

---

**Algorithm 1:** Layer Wise CNN Learning

**Input** : Training samples $\{x_0^n, y^n\}_{n \leq N}$

1 **for** $j \in 0..J - 1$ **do**
2     Apply Eq.(1) to obtain $\{x_j^n\}_{n \leq N}$
3     Initialize $\theta_j, \gamma_j$
4     $(\theta_j^*, \gamma_j^*) = \arg\min_{\theta_j, \gamma_j} \hat{\mathcal{R}}(z_{j+1}; \theta_j, \gamma_j)$
5 **end**

---

At depth $j$, assume we have constructed the parameters $\{\theta_0^*, ..., \theta_j^*\}$. Our algorithm can produce samples $\{x_j^n\}$. Taking $z_{j+1} = z(x_j^n; \theta_j, \gamma_j)$, we will employ an optimization procedure that aims to minimize the risk $\hat{\mathcal{R}}(z_{j+1}; \theta_j, \gamma_j)$. This procedure (Alg. 1) consists in training (e.g. using SGD) the shallow CNN classifier $C_j$ on top of $x_j$, to obtain the new parameter $\theta_{j+1}^*$. Under mild conditions, it improves the training error at each layer as shown below:

**Proposition 3.1** (Progressive improvement). *Assume that $P_j = Id$. Then there exists $\theta_0$ such that:*

$$\hat{\mathcal{R}}(z_{j+1}; \theta_j^*, \gamma_j^*) \leq \hat{\mathcal{R}}(z_{j+1}; \theta_0, \gamma_{j-1}^*) = \hat{\mathcal{R}}(z_j; \theta_{j-1}^*, \gamma_{j-1}^*).$$

*Proof.* As $\rho(\rho(x)) = \rho(x)$, we simply have to chose $\theta_0$ such that $W_{\theta_0} = Id$. $\qquad \square$

A technical requirement for the actual optimization procedure is to not produce a worse objective than the initialization. It can be achieved by taking the best result along the optimization trajectory.

The cascade can inherit from the individual properties of each auxiliary problem. For instance, as $\rho$ is 1-Lipschitz, if each $W_{\theta_j^*}$ is 1-Lipschitz then so is $x_J$ w.r.t. $x$. Another example is the nested objective defined by Alg. 1: The optimality of the solution will be largely governed by the optimality of the sub-problem solver. Specifically, if the auxiliary problem solution is close to optimal than the solution of Alg. 1 will be close to optimal.

**Proposition 3.2.** *Assume the parameters $\{\theta_1^*, ..., \theta_J^*\}$ are obtained via a fixed layerwise optimization procedure. We assume that $W_{\theta_j^*}$ is 1-lipschitz without loss of generality and that the biases are bounded uniformly by $B$. Given an input function $g(x)$, we consider functions of the type $z_g(x) = C_\gamma \rho W_\theta g(x)$. For $\epsilon > 0$, we call $\theta_{\epsilon,g}$ the parameter provided by a procedure to minimize $\hat{\mathcal{R}}(z_g; \theta; \gamma)$ and we assume it finds 1-lipschitz operators that satisfy:*

*1.$\forall g, \tilde{g}, \underbrace{\|\rho W_{\theta_{\epsilon,g}} g(x) - \rho W_{\theta_{\epsilon,\tilde{g}}} \tilde{g}(x)\| \leq \|g(x) - \tilde{g}(x)\|}_{\text{(stability)}}$,   2.$\underbrace{\|W_{\theta_j^*} x_j^* - W_{\theta_{\epsilon, x_j^*}} x_j^*\| \leq \epsilon(1 + \|x_j^*\|)}_{\text{($\epsilon$-approximation)}}$,*

*with, $\tilde{x}_{j+1} = \rho W_{\theta_{\epsilon, \tilde{x}_j}} \tilde{x}_j$ and $x_{j+1}^* = \rho W_{\theta_j^*} x_j^*$ with $x_0^* = \tilde{x}_0 = x$, then, we prove by induction:*

$$\|x_J^* - \tilde{x}_J\| = \mathcal{O}(J^2 \epsilon) \tag{3}$$

The proof can be found in the Appendix A. Thus, we have demonstrated an example of how our training strategy can permit to extend results from shallow CNNs to deeper CNNs, in particular for $k = 1$.

### 3.3 Auxiliary Problems and the Properties They Induce

We now discuss the properties arising from the auxiliary problems. We start with $k = 1$, for which the auxiliary classifier consists of only the linear $A$ and $L$ operators. Thus, the optimization aims to obtain the weights of a 1-hidden layer NN. For this case, as discussed in Sec. 1, a variety of theoretical results exist (e.g. (Cybenko, 1989; Barron, 1994)). Moreover, Arora et al. (2018); Ge et al. (2017); Du & Goel (2018); Bach (2014) proposed provable optimization strategies for this case. Thus the analysis and optimization of the 1-hidden layer problem is a case that is relatively well understood compared to deep counterparts. At the same time, as shown in Prop. 3.2, applying an existing optimization strategy could give us a bound on the solution of the overall objective of Alg. 1. This shows that our training strategy can be more amenable to the development of provable learning for deep CNNs, while our experiment will show it can still yield high performance networks.

Furthermore, for the case of $k = 1$, the optimization of the 1-hidden layer network will encourage the hidden layer outputs to, maximally, linearly separate the training data. Specializing Prop. 3.1 for this case shows that the layerwise $k = 1$ procedure will try to progressively improve the linear separability. Progressive linear separation has been empirically studied in end-to-end CNNs (Zeiler & Fergus, 2014; Oyallon, 2017) as an indirect consequence, while the $k = 1$ training permits us to study this basic principle more directly as the layer objective in the sequel.

Unique to our layer-wise learning formulation, we consider the case where the auxiliary learning problem involves several auxiliary hidden layers. We will interpret, and empirically verify, in Sec. 4.2 that this builds layers that are progressively better inputs to shallow CNNs. We will also show a link to building, in a more progressive manner, linearly separable layers. Considering only shallow (with respect to total depth) auxiliary problems (e.g. $k = 2, 3$ in our work) we can maintains several advantages. Indeed, optimization for shallow networks is generally easier, as we can for example diminish the vanishing gradient problem, reducing the need for identity loops or normalization techniques (He et al., 2016). Two and three layers networks are also appealing for extending results from one hidden layer as they are the next natural member in the family of NNs.

## 4 Experiments and Discussion

We performed experiments on the large-scale ImageNet-1k (Russakovsky et al., 2015), a major catalyst for the recent popularity of deep learning, as well as the CIFAR-10 dataset. We study the classification performance of layerwise models with $k = 1$, comparing them to standard benchmarks and other sequential learning methods. Then we inspect the representations built through our auxiliary tasks and motivate the use of models learned with auxiliary hidden layers $k > 1$, which we subsequently evaluate at scale.

We call $M$ the number of feature maps of the first convolution of the network and $\tilde{M}$ the number of feature maps of the first convolution of the auxiliary classifiers. This fully defines the width of all the layers, since input width and output width are equal unless the layer has downsampling, in which case the output width is twice the input width. Finally, $A$ is chosen to average over the four spatial quadrants, yielding a $2 \times 2$-shaped output. Spatial averaging before the linear layer is common in ResNets (He et al., 2016) to reduce size. In our case this is critical to permit scalability to large image sizes at early layers of layer-wise training. For computational reasons on ImageNet, an invertible downsampling is also applied (reducing the signal to output $12 \times 112^2$). We also construct an ensemble model, which consists of a weighted average of all auxiliary classifier outputs, i.e. $Z = \sum_{j=1}^{J} 2^j z_j$.

We briefly introduce the datasets and preprocessing. The CIFAR-10 dataset consists of small RGB images with $50k$ samples for training and $10k$ samples for testing. We use the standard data augmentation and optimize each layer with SGD using a momentum of $0.9$ and a batch-size of $128$. The initial learning rate is set to $0.1$ and we use the reduced schedule with decays of $0.2$ every $15$ epochs (Zagoruyko & Komodakis, 2016), for a total of 50 epochs in each layer. The ImageNet dataset consists of $1.2M$ RGB images of size varying size for training. Our data augmentations consists of random crops of size $224^2$. At testing time, the image is rescaled to a size of $256^2$ then cropped at size $224^2$. We used SGD with momentum $0.9$ for a batch size of 256. The initial learning rate is $0.1$ (He et al., 2016) and we use the reduced schedule with decays of $0.1$ every 20 epochs for 45 epochs. We use 4 GPUs to train our ImageNet models.

## 4.1 ALEXNET ACCURACY WITH 1-HIDDEN LAYER AUXILIARY PROBLEMS

We consider the, atomic, layerwise CNN with $k = 1$ which corresponds to solving a sequence of 1-hidden layer CNN problems. As discussed in Sec. 2, previous attempts at supervised layerwise training (Fahlman & Lebiere, 1990a; Arora et al., 2014; Huang et al., 2017; Malach & Shalev-Shwartz, 2018), which rely solely on sequential solving of shallow problems have yielded performance well below that of typical deep learning models on the CIFAR dataset. None of them have scaled to datasets such as ImageNet, where end-to-end CNNs have proved absolutely critical (Bartunov et al., 2018). We show, surprisingly, that it is possible to go beyond the AlexNet performance barrier (Krizhevsky et al., 2012) without end-to-end backpropagation on ImageNet with this elementary auxiliary problem. To emphasize the stability of the training process and to permit comparison the original AlexNet architecture we do not apply any batch-norm to this model.

**CIFAR-10.** We trained a model with $J = 5$ layers, down-sampling at layers $j = 1, 3$, and layer sizes starting at $M = 256$. We obtain **88.3%** and note that this accuracy is close to the AlexNet model performance (Krizhevsky et al., 2012) for CIFAR-10 (89.0%). Previous attempts at sequentially trained 1-hidden layer networks have yielded performance that do not exceed that of the top hand-crafted methods or those using unsupervised learning. To the best of our knowledge they obtain $82.0\%$ accuracy (Huang et al., 2017). The end-to-end version of this model obtains $89.7\%$, we note the lack of batch-norm is more detrimental in the end to end setting. Full comparisons are shown in Table 2.

**ImageNet.** Our model is trained with $J = 8$ layers and downsampling operations at layers $j = 2, 3, 4, 6$. Layer sizes start at $M = 256$. Our final trained model achieves **79.7%** top-5 single crop accuracy on the validation set and **80.8%** a weighted ensemble of the layer outputs. In addition to exceeding AlexNet, this model compares favorably to all alternatives to end-to-end supervised CNNs including hand crafted computer vision and unsupervised learning techniques (Noroozi & Favaro, 2016; Perronnin & Larlus, 2015; Sánchez et al., 2013; Oyallon et al., 2017) (full results shown in Table 1). We also note that our final training accuracy is relatively high for ImageNet (87% - see also Appendix B), which indicates that appropriate regularization may lead to a further scaling. We now look at empirical properties induced in the layers and subsequently evaluate the distinct $k > 1$.

## 4.2 EMPIRICAL SEPARABILITY PROPERTIES

We study the intermediate representations generated by the layerwise learning procedure in terms of linear separability as well as separability by a more general set of classifiers. Our aims are **(a)** to determine empirically whether $k = 1$ indeed progressively builds more and more linearly separable data representations and **(b)** to determine how linear separability of the representations evolves for networks constructed with $k > 1$ auxiliary problems. Finally we ask whether the notion of building progressively better inputs to a linear model ($k = 1$ training) has an analogous counterpart for $k > 1$: building progressively better inputs for shallow CNNs (discussed in Sec 3.3).

We define *linear separability* of a representation as the maximum accuracy achievable by a linear classifier. Further we define the notion of *CNN-p-separability* as the accuracy achieved by a $p$-layer CNN trained on top of the representation to be assessed.

We focus on CNNs trained on CIFAR-10 without downsampling. Here, $J = 5$ and we vary the layer sizes $M = 64, 128, 256$. The auxiliary classifier feature map size, when applicable, is $\tilde{M} = 256$. We train with 5 random initializations for each network and report an average standard deviation of 0.58% test accuracy. Each layer is evaluated by training a one-versus-rest logistic regression, as well as $p = 1, 2$-hidden-layer CNN on top of these representations. Because the linear representation has been optimized for it, we spatially average to a $2 \times 2$ shape before feeding them to our learning algorithms. Fig. 2 shows the results of each of these evaluations plotting test set accuracy curves as a function of neural network depth for each of the three evaluations. For these plots we averaged over initial layer sizes $M$ and classifier layer sizes $\tilde{M}$ and random seeds. Each individual curve closely resembles these average curves, with slight shifts in the y-axis, depending on $M$ and $\tilde{M}$.

We observe that linear separability monotonically increases with layer depth as expected from Sec. 3.3 for $k = 1$. Interestingly, we find that linear separability also obtains in the case of $k > 1$, even though it is not directly specified by the auxiliary problem objective. At earlier layers, linear separation

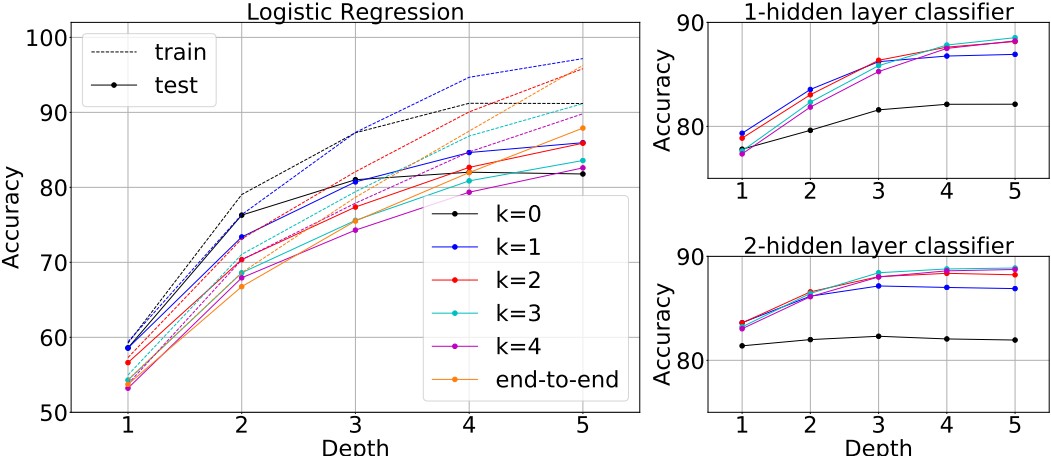

Figure 2: (Left) Linear and (Right) CNN-$p$ separability as a function of depth for CIFAR-10 models. For Linear separability we aggregate across $M = 64, 128, 256$, individual results are shown in Appendix C.2, the relative trends are largely unchanged, although overall accuracies are higher in larger $M$. For CNN-$p$ probes, all models achieve $100\%$ train accuracy at the first or 2nd layer, thus only test accuracy is reported.

capability of models trained with $k = 1$ increases fastest as a function of layer depth compared to models trained with deeper auxiliary networks, but flattens out to a lower asymptotic linear separability at deeper layers. This shows that the simple principle of the $k = 1$ objective that tries to produce the maximal linear separation at each layer might not be an optimal strategy for achieving "progressive" linear separation.

We also notice that the deeper the auxiliary classifier, the slower is the increase in linear separability initially, but the higher is the linear separability at deeper layers. From the two right diagrams we also find that the CNN-$p$-separability progressively improves - but much more so for $k > 1$ trained networks. This shows that linear separability of a layer is not the sole criterion for rendering a representation a good "input" for a CNN. It further shows that our sequential training procedure for the case $k > 1$ can indeed build a representation that is progressively a better input to a shallow CNN.

### 4.3    Scaling up Layerwise CNNs with 2 and 3 Hidden Layer Auxiliary Problems

We study the training of deep networks with $k = 2, 3$ hidden layer auxiliary problems. We limit ourselves to this setting to keep the auxiliary models shallow with respect to the network depth. We employ widths of $M = 128$ and $\tilde{M} = 256$ for both CIFAR-10 and ImageNet. For CIFAR-10, the total number of layers is $J = 4$. A downsampling is applied at depth $j = 2$. For ImageNet we closely follow the VGG architectures, which with their $3 \times 3$ convolutions and absence of skip-connections bear strong similarity to ours. We use $J = 8$, giving 11 total layers (similar to e.g. VGG-11). As we start at halved resolution we do only 3 downsamplings at $j = 2, 4, 6$. Unlike the $k = 1$ case we found it helpful to employ batch-norm for these auxiliary problems.

We report our results for $k = 2, 3$ in Table 2 (CIFAR-10) and Table 1 (ImageNet) along with the results for our $k = 1$ model. As expected from the previous section, the transition from $k = 1$ to $k = 2, 3$ improves the performances substantially. We compare our CIFAR-10 results to other sequentially trained propositions in the literature. Our methods exceed these in performance by a large margin, while the ensemble model of $k = 3$ surpasses the VGG, the other sequential models perform do not exceed unsupervised methods. No alternative sequential models are available for ImageNet. We thus compare our results on ImageNet to the standard reference CNNs and the best-performing alternatives to end-to-end Deep CNNs. Our $k = 3$ layerwise ensemble model achieves **89.8%** accuracy, which is comparable to VGG-13 and largely exceeds AlexNet performance. The reference model accuracies for AlexNet, VGG, and ResNet-152 use the same input sizes and single

| Layerwise Trained | Top-1 (Ens.) | Top-5 (Ens.) |
|---|---|---|
| Layerwise, $k = 1$ | 58.1 (59.3) | 79.7 (80.8) |
| Layerwise $k = 2$ | 65.7 (67.1) | 86.3 (87.0) |
| Layerwise $k = 3$ | 69.7 (**71.6**) | 88.7 (**89.8**) |
| Layerwise $k = 3$, $\tilde{M}_f = 1024$ | 69.2 | 88.6 |
| Layerwise $k = 3$, $\tilde{M}_f = 512$ | 68.7 | 88.5 |
| **End-to-End Deep CNN** | | |
| AlexNet | 56.5 | 79.1 |
| VGG-11 | 69.0 | 88.6 |
| VGG-13 | 69.9 | 89.3 |
| VGG-19 | 72.9 | 90.9 |
| Resnet-152 | **78.3** | **94.1** |
| End-to-end of $k = 3, \tilde{M}_f = 512$ | 71.5 | 90.1 |
| **Alternatives** | | |
| Unsup + MLP (Noroozi et al, 2016) | 34.6 | N/A |
| FV+ MLP (Perronin et al., 2015) | **55.6** | **78.4** |
| FV + SVM (Sánchez et al., 2013) | 54.3 | 74.3 |

| Layer-wise Trained | Acc. (Ens.) |
|---|---|
| Layerwise $k = 1$ | 88.3 (88.4) |
| Layerwise $k = 2$ | 90.4 (90.7) |
| Layerwise $k = 3$ | 91.7 (**92.8**) |
| BoostResnet. (Huang et al., 2017) | 82.1 |
| ProvableNN (Malach et al., 2018) | 73.4 |
| (Mosca et al., 2017) | 81.6 |
| **End-to-End Deep CNN** | |
| AlexNet | 89 |
| VGG [1] | 92.5 |
| WRN 28-10 (Zagoruyko et al. 2016) | **96.0** |
| End-to-end $k$=1 | 89.7 |
| **Alternatives** | |
| (Oyallon & Mallat, 2015) | 82.3 |
| Unsup. + SVM (Dosovitskiy et al., 2014) | **84.3** |

Table 1: Single crop validation acc. on ImageNet. Our models use $J = 8$. In parentheses see the ensemble prediction. $\tilde{M}_f$ specifies the auxiliary network for models that have the final auxiliary network replaced. These show minor loss to the original bigger auxiliary. Layer-wise models are competitive with benchmarks that similarly don't use skip connections and outperform all other alternatives to end-to-end.

Table 2: Results on CIFAR-10. Compared to the few existing methods using *only* layerwise training schemes we report substantial performance improvement. Overall our models are competitive with well known benchmarks models that like ours do not use skip connections.

crop evaluation[1]. Reference models relying on residual connections and very deep networks have substantially better performance than our models. We believe that one can extend layer-wise learning to these modern techniques. However, this is outside the scope of this work. Moreover, recent ImageNet models (after VGG) are developed in industry settings, with large scale infrastructure available for architecture and hyper-parameter search.

We emphasize that our approach enables the training of much larger layerwise models than end-to-end ones on the same hardware. This suggests applications in fields with large models (e.g. 3-D vision and medical imaging). We also observed that using outputs of early layers that were not yet converged still permitted improvement in subsequent layers. This suggests that our framework might allow an extension that solves the auxiliary problems in parallel to a certain degree.

**Reducing final auxiliary network** Recall $\tilde{M}$ is the width of the initial auxiliary CNN. Let $\tilde{M}_f$ denote the width of the final auxiliary CNN. In the experiments above this is relatively large ($\tilde{M}_f = 2048$). We observed that although a larger $\tilde{M}$ during training can be beneficial, particularly in earlier layers, the final representation will tend to be a good input even for smaller classifier networks fit after the primary network (up to $x_{J-1}$) is trained. We thus reduce the width of the final *auxiliary* network $k = 3$ by performing an extra, smaller auxiliary problem evaluation to update step $j = 7$. We use auxiliary networks of size $\tilde{M}_f = 512$ and $1024$ (instead of $2048$). While the model size is reduced substantially, we observe only a limited loss of accuracy. For comparison, we train an end-to-end network with the same architecture as our $J = 8$ network with the final auxiliary of $\tilde{M}_f = 512$. Tab. 1 shows the accuracy of the end-to-end model is $90.1\%$ top-5 compared to $88.5\%$ top-5 for the sequentially trained. We show a direction to close this relatively small gap in the next section.

---

[1] Accuracies are reported from the tables in http://torch.ch/blog/2015/07/30/cifar.html and https://pytorch.org/docs/master/torchvision/models.html. Our code is based on the same software package.

**Layerwise Model Compression**   Wide, overparametrized, layers have been shown to be important for learning(Neyshabur et al., 2018), but it is often possible to reduce the layer size *a posteriori* without losing significant accuracy  (Hinton et al., 2014; LeCun et al., 1990). For the specific case of CNNs, one technique removes channels heuristically and then fine-tunes(Molchanov et al., 2016).In our setting, a natural strategy presents itself, which integrates compression into the learning process: **(a)** train a new layer (via an auxiliary problem) and **(b)** immediately apply model compression to the new layer. The model-compression-related fine-tuning operates over a single layer, making it fast and the subsequent training steps have a smaller input and thus fewer parameters, which speeds up the sequential training. We implement this approach using the filter removal technique of Molchanov et al. (2016) only at each newly trained layer, followed by a fine-tuning of the auxiliary network. We test this idea on CIFAR-10. A baseline network of 5 layers of size $64$ (no downsampling, trained for 120 epochs and lr drops each 25 epochs) obtains an end-to-end performance of $87.5\%$. We use our layer-wise learning with $k = 3, J = 3, M = 128, \tilde{M} = 128$. At each step we prune each layer from $128$ to $64$ filters and subsequently fine-tune *the auxiliary network* to the remaining features over 20 epochs. We then use a final auxiliary of $\tilde{M}_f = 64$ obtaining a sequentially learned, final network of the same architecture as the baseline. The final accuracy is $87.6\%$, which is very close to the baseline. We note that each auxiliary problem incurs minimal reduction in accuracy through feature reduction. Unlike the previous experiment, where our final performance was slightly below that of end-to-end on the same architecture, this gap could be closed by easy-to-integrate compression approaches.

## 5   CONCLUSION

We have shown, to the best of our knowledge, the first alternative to end-to-end learning that scales on large-scale benchmarks such as ImageNet and can be competitive with standard CNN baselines. We build competitive models by training only shallow CNNs and using standard architectural elements (ReLU, convolution). This shows that the approach is generic and could be adapted to more complex classes of NNs. Layerwise training opens the door to applications such as larger models under memory constraints, model prototyping, joint model compression and training, and more stable training for challenging scenarios. The framework may be extendable to the parallel training of layers as well as the development of novel localized feedback mechanisms. Importantly, our results suggest a number of open questions regarding the mechanisms that underlie the success of CNNs: for example can the 1-hidden layer network objective be better specified, filling in the gap between 1-hidden layer network and the $k > 1$ approach? Moreover, our models can potentially provide easier to study, high performance models for researchers.

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

## A  PROOF OF PROPOSITION

**Proposition A.1.** *Assume the parameters $\{\theta_1^*, ..., \theta_J^*\}$ are obtained via a fixed layerwise optimization procedure. We assume that $W_{\theta_j^*}$ is 1-lipschitz without loss of generality and that the biases are bounded uniformly by $B$. Given an input function $g(x)$, we consider functions of the type $z_g(x) = C_\gamma \rho W_\theta g(x)$. For $\epsilon > 0$, we call $\theta_{\epsilon, g}$ the parameter provided by a procedure to minimize $\hat{\mathcal{R}}(z_g; \theta; \gamma)$ and we assume it finds 1-lipschitz operator that satisfy:*

*1.$\forall g, \tilde{g}, \underbrace{\|\rho W_{\theta_{\epsilon,g}} g(x) - \rho W_{\theta_{\epsilon,\tilde{g}}} \tilde{g}(x)\| \leq \|g(x) - \tilde{g}(x)\|}_{\text{(stability)}},$  2.$\underbrace{\|W_{\theta_j^*} x_j^* - W_{\theta_{\epsilon,x_j^*}} x_j^*\| \leq \epsilon(1 + \|x_j^*\|)}_{\text{($\epsilon$-approximation)}},$*

*with, $\tilde{x}_{j+1} = \rho W_{\theta_{\epsilon,\tilde{x}_j}} \tilde{x}_j$ and $x_{j+1}^* = \rho W_{\theta_j^*} x_j^*$ with $x_0^* = \tilde{x}_0 = x$, then, we prove by induction:*

$$\|x_J^* - \tilde{x}_J\| = \mathcal{O}(J^2 \epsilon) \tag{4}$$

*Proof.* First observe that $\|x_{j+1}^*\| \leq \|x_j^*\| + B$ by non expansivity. Thus, by induction, $\|x_j^*\| \leq jB + \|x\|$. Then, let us show that: $\|x_j^* - \tilde{x}_j\| \leq \epsilon(\frac{j(j-1)}{2} B + j\|x\| + j)$ by induction. Indeed, for $j + 1$:

$$
\begin{aligned}
\|x_{j+1}^* - \tilde{x}_{j+1}\| &= \|\rho W_{\theta_j^*} x_j^* - \rho W_{\theta_{\epsilon,\tilde{x}_j}} \tilde{x}_j\| \\
&= \|\rho W_{\theta_j^*} x_j^* - \rho W_{\theta_{\epsilon,x_j^*}} x_j^* + \rho W_{\theta_{\epsilon,x_j^*}} x_j^* - \rho W_{\theta_{\epsilon,\tilde{x}_j}} \tilde{x}_j\| \\
&\leq \|W_{\theta_j^*} x_j^* - W_{\theta_{\epsilon,x_j^*}} x_j^*\| + \|\rho W_{\theta_{\epsilon,x_j^*}} x_j^* - \rho W_{\theta_{\epsilon,\tilde{x}_j}} \tilde{x}_j\| \text{ by non-expansivity} \\
&\leq \epsilon\|x_j^*\| + \epsilon + \|x_j^* - \tilde{x}_j\| \text{ from the assumptions} \\
&\leq \epsilon(jB + \|x\| + 1) + \|x_j^* - \tilde{x}_j\| \text{ from above} \\
&\leq \epsilon(jB + \|x\| + 1) + \epsilon(\frac{j(j-1)}{2} B + j + j\|x\|)\| \text{ by induction} \\
&= \epsilon(\frac{j(j+1)}{2} B + (j+1)\|x\| + (j+1))
\end{aligned}
$$

As $x_0^* = x_0^\epsilon$, the property is true for $j = 0$.

□

## B  ADDITIONAL DETAILS ON IMAGENET MODELS AND PERFORMANCE

For ImageNet we report the improvement in accuracy obtained by adding layers in Figure 3 as seen by the auxiliary problem solutions. We observe that indeed the accuracy of the model on both the training and validation is able to improve from adding layers as discussed in depth in Section 4.2. We observe that $k = 1$ also over-fits substantially.

We provide a more explicit view of the network sizes in Table 3 and Table 4. We also show the number of parameters in the ImageNet networks in Table 6. Although some of the models are not as parameter efficient compared to the related ones in the literature, this was not a primary aim of the investigation in our experiments and thus we did not optimize the models for parameter efficiency (except explicitly at the end of Sec. 4.3), choosing our construction scheme for simplicity. We highlight that this is not a fundamental problem in two ways: (a) for the $k = 1$ model we note that removing the last two layers reduces the size by $1/4$, while the top 5 accuracy at the earlier J=6 layer is 78.8 (versus 79.7), see Figure 3 for detailed accuracies. (b) Our models for $k = 2, 3$ have most of their parameters in the auxiliary network which is easy to correct for once care is applied to this specific point as at the end of Sec. 4.3. The final model we use to compare greedy layerwise training to end-to-end training, $k = 3, M_f = 512$, is actually more parameter efficient than those in the VGG family while having similar performance. We also point out that we use for simplicity the VGG style

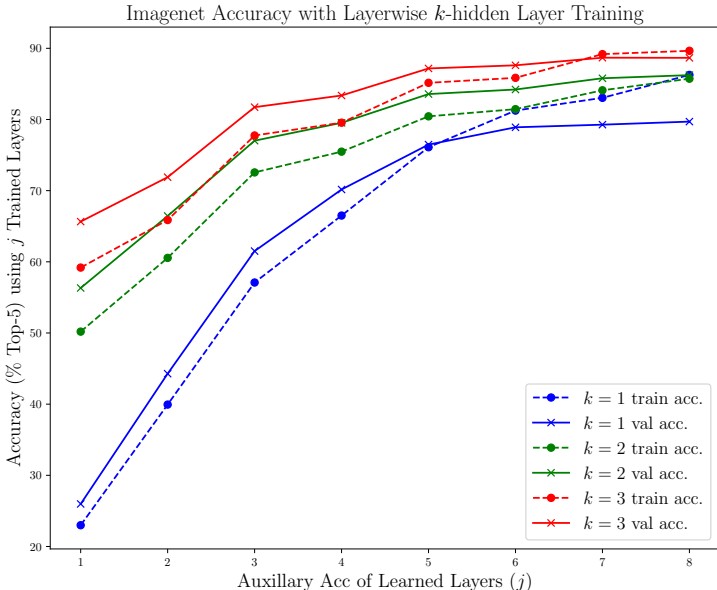

Figure 3: Intermediate Accuracies of models in Sec. 4.3. We note that the $k = 1$ model is larger than the $k = 2, 3$ models.

| Layer | spatial size | layer output size |
|---|---|---|
| Input | $112 \times 112$ | 12 |
| 1 | $112 \times 112$ | 128 |
| 2 | $112 \times 112$ | 128 |
| 3 | $56 \times 56$ | 256 |
| 4 | $56 \times 56$ | 256 |
| 5 | $28 \times 28$ | 512 |
| 6 | $28 \times 28$ | 512 |
| 7 | $14 \times 14$ | 1024 |
| 8 | $14 \times 14$ | 1024 |

Table 3: Network structure for $k = 2, 3$ imagenet models, not including auxiliary networks. Note an invertible downsampling is applied on the input 224x224x3 image to producie the initial input. The default auxillary networks for both have $\tilde{M}_f = 2048$ with 1 and 2 auxiliary layers, respectively. Note auxillary networks always reduce the spatial resolution to $2x2$ before the final linear layer.

construction involving only $3x3$ convolutions and downsampling operations that only half the spatial resolution, which indeed has been shown to lead to relatively less parameter efficient architectures He et al. (2016), using less uniform construction (larger filters and bigger pooling early on) can yield more parameter efficient models.

## C  ADDITIONAL STUDIES

We report additional studies that elucidate the critical components of the system and demonstrate the transferability properties of the greedily learned features.

### C.1  CHOICE OF DOWNSAMPLING

In our experiments we use primarily the invertible downsampling operator as the downsampling operation. This choice is to reduce architectural elements which may be inherently lossy such as average pooling. Compared to maxpooling operations it also helps to maintain the network in Sec. 4.1 as a pure ReLU network, which may aid in analysis as the maxpooling introduces an additional

| Layer | spatial size | layer output size |
|-------|--------------|-------------------|
| Input | $112 \times 112$ | 12 |
| 1 | $112 \times 112$ | 256 |
| 2 | $112 \times 112$ | 256 |
| 3 | $56 \times 56$ | 512 |
| 4 | $28 \times 28$ | 1024 |
| 5 | $14 \times 14$ | 2048 |
| 6 | $14 \times 14$ | 2048 |
| 7 | $7 \times 7$ | 4096 |
| 8 | $7 \times 7$ | 4096 |

Table 4: Network structure for $k = 1$ ImageNet models, not including auxiliary networks. Note an invertible down-sampling is applied on the input 224x224x3 image to produce the initial input. Note this network does not include any batch-norm.

| Layer-wise Trained | Acc. (Ens.) |
|--------------------|-------------|
| Strided Convolution | 87.8 |
| Invertible Down | 88.3 |
| AvgPool | 87.6 |
| MaxPool | 88.0 |

Table 5: Comparison of different downsampling operations

| Models | Number of Parameters |
|--------|----------------------|
| Our model $k = 3$, $M_f = 512$ | 46M |
| Our model $k = 3$ | 102M |
| Our model $k = 2$ | 64M |
| Our model $k = 1$, $J = 8$ | 406M |
| Our model $k = 1$, $J = 6$ | 96M |
| AlexNet | 60M |
| VGG-16 | 138 M |

Table 6: Overall parameter counts for imagenet models trained in Sec. 4 and from literature.

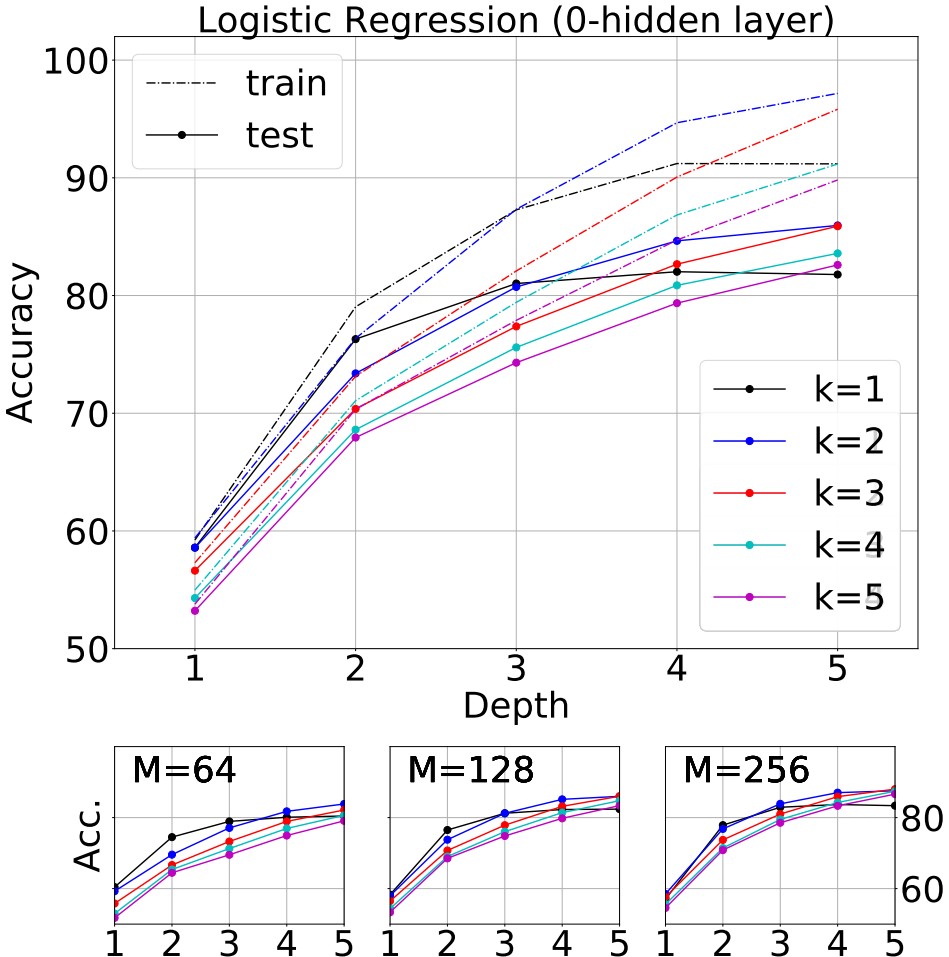

Figure 4: Linear separability of differently trained sequential models. We show how the data varies for the different $M$, observing similar trends to the aggregated data.

non-linearity. We show here the effects of using alternative downsampling approaches including: average pooling, maxpooling and strided convolution. On the CIFAR dataset in the setting of $k = 1$ we find that they ultimately lead to very similar results with invertible downsampling being slightly better. This shows the method is rather general. In our experiments we follow the same setting described for CIFAR. The setting here uses $J = 5$ and downsamplings at $j = 1, 3$. The size is always halved in all cases and the downsampling operation and the output sizes of all networks are the same. Specifically the Average Pooling and Max Pooling use $2 \times 2$ kernels and the strided convolution simply modifies the $3 \times 3$ convolutions in use to have a stride of 2. Results are shown in Table 5.

## C.2 EFFECT OF WIDTH

We report here an additional view of the aggregated results for linear separability discussed in Sec. 4.2. We observe that the trend of the aggregated diagram is similar when comparing only same sized models, with the primary differences in model sizes being increased accuracy.

We perform an ablation to demonstrate the effect of width on the imagenet dataset. We show results for the model $k = 3$ from Sec. 4.3 with all layer sizes halved. Results are reported for all auxiliary models in Table 7, we note our results are consistent with similar studies of width from Zagoruyko & Komodakis (2016) showing there is some gains from wider layers (e.g. 2% top 5) but emphasizing that the effect of the cascade is completely critical to obtaining high performance.

| layer | 1 | 2 | 3 | 4 | 5 | 6 | 7 | 8 |
|---|---|---|---|---|---|---|---|---|
| $M = 128$ | 65.7 | 71.9 | 81.7 | 83.4 | 87.2 | 87.5 | 88.5 | 88.7 |
| $M = 64$ | 60.2 | 69.0 | 76.6 | 77.8 | 82.9 | 83.7 | 86.2 | 86.8 |

Table 7: For $k = 3$ model we report the effect of width. We compare halving the size of the model and the accuracy at each layer we report the accuracy of the auxiliary model.

| | Accuracy |
|---|---|
| ConvNet from scratch (Zeiler & Fergus, 2014) | $46 \pm 1.7\%$ |
| Layer 1 | $45.5 \pm 0.9$ |
| Layer 2 | $59.9 \pm 0.9$ |
| Layer 3 | $70.0 \pm 0.9$ |
| Layer 4 | $75.0 \pm 1.0$ |
| Layer 8 | $82.6 \pm 0.9$ |

Table 8: Accuracy obtained by a linear model using the features of the $k = 1$ network at a given layer on the Caltech-101 dataset. We also give the reference accuracy without transfer.

## C.3 TRANSFER LEARNING ON CALTECH-101

Deep CNNs such as AlexNet trained on Imagenet are well known to have generic properties for computer vision tasks, permitting transfer learning on many downstream applications. We briefly evaluate here if the $k = 1$ imagenet model (Sec. 4.1) shares this generality on the Caltech-101 dataset. This dataset has 101 classes and we follow the same standard experimental protocol as Zeiler & Fergus (2014): 30 images per class are randomly selected, and the rest is used for testing. The average per class accuracy is reported using 10 random splits. As in Zeiler & Fergus (2014) we restrict ourselves to a linear model. We use a multinomial logistic regression applied on features from different layers including the final one. For the logistic regression we rely on the default hyperparameter settings for logistic regression of the `sklearn` package using the SAGA algorithm. We apply a linear averaging and PCA transform (for each fold) to reduce the dimensionality to 500 in all cases. We find the results are similar to those reported in Zeiler & Fergus (2014) for their version of the AlexNet. This highlights the model has similar transfer properties and also shows similar progressive linear separability properties as the end-to-end trained AlexNet.

