# OpenReview forum: "Shallow Learning For Deep Networks"
_ICLR.cc/2019/Conference_

### Official Review · AnonReviewer3 · 2018-10-31
**Good paper but it is not clear what is the main ingredient behind its success**

**Rating:** 7
**Confidence:** 4

**Review:**

Summary:

This paper proposes layer wise training of neural networks using classification auxiliary tasks for training each layer. Experiments are presented on CIFAR10 and Imagenet. Accuracies close to end to end training are obtained.

The layer wise training is repeated for J steps, the auxiliary tasks are trained on top of the shallow one layer (of width M ) with a network  of depth k and width tilde{M}. Layerwise training is done using sgd with momentum, and the learning rate is decayed through epochs.

Note that the layer wise training is done with large width M than typical end to end networks in use.

The authors argue and test the hypothesis that auxiliary tasks  encourage the linear separability of CNN features.

To reduce the size of the learned network the author propose a layer wise compression using the filter removal technique of Molchanov et al .

Reproducibility:

This empirical  work has been investigated for a while with mild success, the authors should make their code available to the community to confirm and reproduce  their findings.  I encourage the authors to make their code available during the review/discussion period.


Significance of the work:

From reading the paper it is not clear what is the main ingredient that makes this layer wise training  successful, negative results would help in understanding what is important for the accuracy.

Some more ablation studies and negative results will be insightful and here are few suggestions in that direction:

- Authors claim that they used invertible downsampling as max or average pooling  lead to information loss. Does the layer wise training give worst results with average or max pooling? If so please report those numbers to know what is the implications of this choice of pooling.

- On the width of the networks, seems it is key for the success of the approach.  What if  you train wider networks with J that is small? (  J=3 for instance but much  wider networks, instead of J=8 now for imagenet.)

- To answer the same question above one needs also to see what are the accuracies for J=8 with thiner networks (smaller M )?

- Would the accuracy  with the layer wise training reach a  plateau if one uses an architecture with J higher than 8?

- Transferability of the learned features: end to end features are know to be transferable. It would be good to see if this still holds using the network layer wise trained on imagenet for CIFAR10 or other datasets.

Other Questions:
- Section 3.2 is vague. In Proposition 3.1 and  Proposition 3.2 can you add some text to explain what are the implicitions of the claims? “Thus our training permits to extend some results on shallow CNN to deeper CNNs …” which shallow results ?

- “For k>1 batch normalization was useful “ is this only on the auxiliary problems networks  or you used also batch norm for the layer ?

- The ensemble used is Z=\sum_{j=1}^J 2^j z_j , this uses the network of J layers ,  also the O(J) auxiliary networks  of depth k. Please report the number of parameters for all models (single and ensembles) in Table 1 and Table 2.

- In the conclusion: “The framework can be extendable to the parallel training …” how would this possible since one needs the output of the first training to do the training of the next layer. can you elaborate on what is meant here?

Minor:
page 2 bottom have competitorsand -> have competitors and
the non linearity rho in equation 1 and throughout the paper put a bracket for its argument \rho(x) not \rho x
Page 6 , Imagenet paragraph : W —> We
section 4.2 we define linear separability etc… a space is missing before Further
section 4.3 we report our results .. (Imagenet) a space is missing after ImageNet)

Overall Assessment:
This is a good paper, making the code available and adding more ablation studies and explanations of width versus depth and the choice of pooling will make the contributions easier to understand.

---

> ### Author Response · Authors · 2018-11-11
> **Response to R3 [1/2]**
>
> Thank you for your positive review. We agree that our work raises a lot of interesting questions, and we have added several appendices to address your concerns. We address the points sequentially below:
>
> << I encourage the authors to make their code available during the review/discussion period. >>
> As requested we have released an anonymized code repository: https://anonymous.4open.science/repository/75115ffe-d110-4814-ade0-060dd12b9f7e/
>
> Please note this is a very preliminary release created as per request and we plan to have a more clean code at a later date.
>
> <<From reading the paper it is not clear what is the main ingredient that makes this layer wise training  successful>>
> We believe the most critical, yet simple, engineering ingredient to the scalability is the spatial averaging operation applied as the first part of the fc in auxiliary classifier. For example even in k=1 without this operation the sizes of the auxillary layers would explode for imagenet-size images (224x224).  We have included some additional ablations in the appendix based on your questions.
>
> <<Does the layer wise training give worst results with average or max pooling?>>
> Thanks for pointing this out, in the text we did not say the max or average pooling is worse, but can see how this might sound implied. The choice of invertible downsampling is driven by the idea of eliminating as many factors as possible that might conceivably be incompatible with greedy learning.  One of them that we identified early on was potential loss of target information (note t that this is an issue hypothesized as a problem with such an approach in (Bengio et al 2006)). Indeed we never established a clear difference in preliminary testing, but still want the inherent lossiness of average pooling and the slightly more complex analytical structure of max pooling to not be a potential factor in our experiments.  We have now completed a more extensive set of ablations included in the revision in Appendix C.1 which shows that, at least for the case of the CIFAR dataset, strided convolutions, max-pooling, and average pooling as downsampling operations yield similar results to invertible downsampling. Invertible downsampling performs marginally better. This shows that the method is rather generic and also contradicts some of the concerns in difficulty to maintain target information in (Bengio et al 2006).
>
>
> <<On the width of the networks, seems it is key for the success of the approach.  What if  you train wider networks with J that is small? (  J=3 for instance but much  wider networks, instead of J=8 now for imagenet.) To answer the same question above one needs also to see what are the accuracies for J=8 with thiner networks (smaller M)?>>
>
> As in typical CNNs the width increases can improve performance to some (relatively small with diminishing returns) degree, see e.g. (Zagoruyko et al, 2016) where increases of 2% top 5 on imagenet are observed for high end models when doubling width. However, we emphasize that the width alone is not sufficient to explain the observed results. Take the example of the extreme cases of the k=1 model where the 1st layer obtains only 23% top 5 accuracy: it seems unlikely that simply increasing width for these shallow cases can achieve the desired accuracies.
>
> We point out several results from the existing appendix and a new ablation added in C.2 to address this point:
> - We have results that demonstrate the change in width for CIFAR in Figure 4 of the appendix showing that larger M can improve results, but not in an extreme fashion.
> -  We also refer the reviewer to examine the k=1 accuracy progression in Figure 3 which shows how the accuracy increases nearly linearly at the early layers.
> - We include an ablation that shows the k=3 network described in Sec 4.3 with the widths halved: as predicted from (Zagoruyko et al, 2016) the accuracy changes from 88.7 to 86.8 -- a rather modest change compared to the effect of the cascade we can derive from Figure 3. This effect would be most pronounced if comparing the per-layer widths of k=1 , we will try to do this in a future revision but currently are limited in resources to run ablations on imagenet.
> - Finally we bring the reviewers’ attention to the fact the receptive field for very low depths might be too small with respect to the full image. Typical cnn designs attempt to cover a substantial part or all of the input image in the receptive field.
>
> <<Would the accuracy  with the layer wise training reach a  plateau if one uses an architecture with J higher than 8?>>
> There are certainly diminishing returns. As shown in  Figure 3 in Appendix the accuracies for k=2,3 plateau around J=8. For completeness we have computed an additional layer for the k=2 imagenet model to further extend this chart (J=9). The result is 86.5 compared to the previous layers 86.3 . Note that end-to-end networks also certainly have significantly diminishing returns for more stacked layers.

---

> ### Author Response · Authors · 2018-11-11
> **Response to R3 [2/2]**
>
> <<Transferability of the learned features: end to end features are know to be transferable. It would be good to see if this still holds using the network layer wise trained on imagenet for CIFAR10 or other datasets. >>
> We have added an appendix C.3 as requested which performs a standard experiment of using imagenet as a feature extractor for Caltech-101. Existing work has shown that this performs drastically better than training a CNN from scratch and we largely confirm this with similar results to those obtained in ( Zeiler & Fergus  , 2013).
>
> << In Proposition 3.1 and  Proposition 3.2 can you add some text to explain what are the implications of the claims? “Thus our training permits to extend some results on shallow CNN to deeper CNNs …” which shallow results ?>>
> Please see the response to R1 regarding the propositions. Prop 3.1 shows that with an appropriate initialization and in the specific case of the ReLU non-linearity one should not decrease the performance of the network on the training set. This is a basic question one might ask to start understanding the layerwise learning. Note that this result does not say what is the rate of improvement. We empirically observe this to be surprisingly rapid for imagenet (see Figure 3) and it is fundamentally one of the highly interesting result of this work, requiring further study and explanation by the community. Prop 3.2 supports our arguments that theoretical results for shallow sub-problems can be easily extended to ones about the cascaded deep network. We do this by explicitly showing it for the case of optimization: an optimality condition on the sub-problem is shown here to yield an optimality condition on the layerwise learning optimization. We note that there is growing literature on 1-hidden layer problem optimization and many of those results can be quite more easily extended in this setting than in end-to-end. We have clarified this in the revision
>
> <<“For k>1 batch normalization was useful “ is this only on the auxiliary problems networks  or you used also batch norm for the layer ?>> We used it on the whole network.
>
> <<“The framework can be extendable to the parallel training …” how would this possible since one needs the output of the first training to do the training of the next layer. can you elaborate on what is meant here?>>
> In short, one could use the output of a not fully trained layer. Note also the brief remark at the bottom of page 7 and beginning of page 8. To detail the observation mentioned there, preliminary investigation shows that if layer 1 is trained even for very few iterations instead of until convergence before being fixed, the subsequent cascade built on top of this layer will be substantially better than if the layer 1 was not updated at all from random initialization. This indicates the layers may not need to be fully optimized to provide very useful signal to those above, suggesting that they can pass useful signals to upper layers while continuing to update simultaneously their parameters based on their own local problems. We believe a detailed discussion of this is well outside the scope and message for this work, and we do not use this observation in our experiments here, training each layer the exact same number of iterations as described and purely sequentially.
>
>
> We have addressed your minor comments. Please note however that while \rho(x) is a real valued function, when  using \rho in Equation (1), we consider it an operator and we use the notation from e.g. https://arxiv.org/pdf/1601.04920.pdf preferring to maintain this standard.

---

> ### Author Response · Authors · 2018-11-27
> **Additional concerns**
>
> Dear Reviewer,
> Thank you again for your review. Please let us know if we have addressed your concerns and whether there is still any major outstanding point you would like clarified.

---

> > ### Comment · AnonReviewer3 · 2018-12-05
> > **reply to author feedback**
> >
> > Thank you for making the code available.  You have addressed my concerns.
> >
> > Transfer learning experiments are nice to see.  The ablation study on the pooling using in appendix C.1 are worth a mention in the main paper when you say:" We decided against strided pooling, average pooling, and the non-linear max-pooling, because these strongly encourage a loss of information", for now this can be a misleading statement.
> >
> >  I think the empirical  work  in this paper  is comprehensive and interesting for layerwise supervised training of neural networks and will bring some discussion to ICLR, and hence I maintain my score.
> >
> >  It would be interesting to see if using similar architectures, if one replaces training with back-propagation for one layer , with ADMM for instance as in https://arxiv.org/pdf/1605.02026.pdf or with least square type training as in https://arxiv.org/pdf/1212.5921.pdf  if those positive results will hold. Training a single layer in ADMM can be made very efficient.  The distributed way you proposing here shares a similarity with the papers mentioned above (ADMM and nested deep networks).

---

> > > ### Author Response · Authors · 2018-12-13
> > > **Reply**
> > >
> > > Thank you for your overall positive review and suggestions. We will revise the sentence indicated. We agree replacing the single layer training with a more efficient method is an interesting direction and thank you for the references.

---

### Official Review · AnonReviewer1 · 2018-11-01
**decent experiments, limited novelty**

**Rating:** 5
**Confidence:** 4

**Review:**

This paper is of reasonable quality and clarity, rather modest originality, perhaps considerable significance in some applications.

Strengths:
- I think this kind of method could be useful for data of very high dimensionality, when it is not possible to train everything end to end.
- The experiments seem to be conducted correctly.
- The paper is well written.

Weaknesses:
- (minor) Abstract: it's kind of funny to say that CIFAR-10 is a large scale image recognition problem.
- What the authors are proposing is quite similar to Lee et al. [1], which was not mentioned in the paper as well as a wide range of papers, which were mentioned. I think it is kind interesting for people to revise these techniques from 10 years ago, but this method is just not that novel.
- The authors highlight that their goal is not using this method as a pre-training strategy, but it would be interesting to see whether it would indeed work better if after the layer-wise training, the whole network would be trained end-to-end.
- Maths in this paper is mostly decorative.
- When comparing different models or training methods (e.g. layer-wise trained AlexNet and end-to-end trained AlexNet), it would make sense to do some hyperparameter search. It is very risky to conclude anything otherwise.
- I would like to see a wall clock time comparison between this and end-to-end training.

[1] Lee et al. Deeply-Supervised Nets. 2015.

---

> ### Author Response · Authors · 2018-11-11
> **Response to R1 [1/2]**
>
> Dear Reviewer 1,
>
> We would like to  thank you very much for your review and time. We answer to your points below,
>
> <<(minor) Abstract: it's kind of funny to say that CIFAR-10 is a large scale image recognition problem>>
>
> We agree with you. This was a typo aimed to simplify the sentence flow and we have corrected this statement in the uploaded revision.
>
> <<What the authors are proposing is quite similar to Lee et al. [1], which was not mentioned in the paper as well as a wide range of papers, which were mentioned.>>
>
> Thank you for the reference - we added it in the revision.
> While there is some similarity in the final overall architecture using an auxiliary objective, Note that they use a completely different training technique (end to end) and their contribution bears little resemblance overall to our work.
> Indeed, the model in Lee et al. is trained end-to-end and the contribution is aimed at stabilizing/guiding end-to-end training. Our work focuses on the viability of a  training method that utilizes individually solved auxiliary problems only.  Please also see the extensive literature review which highlights major differences to our work from many other potentially related ones.
>
> <<I think it is kind interesting for people to revise these techniques from 10 years ago, but this method is just not that novel.>>
>
> We agree that the high-level technique has similarities to a few specific existing works, mainly (Huang et al. (2017), Bengio et al. (2006)) . Because our goal here was not to propose an extremely specific technique but to test the assumption that end-to-end learning of layers is essential, we consider it useful and interesting that we were able to achieve our result with a relatively common approach. We believe that our -  counterintuitive -  numerical results alone represent a major contribution to the literature and are a significant novelty. As a more minor but still substantial contribution, there has not been any scalable (even just computationally speaking) greedy layerwise method in the literature for CNNs. Thus even the specific proposed method has some novel aspects. Please also see our general comment discussion on this point (above).
>
>
> <<The authors highlight that their goal is not using this method as a pre-training strategy, but it would be interesting to see whether it would indeed work better if after the layer-wise training, the whole network would be trained end-to-end.>>
>
> We think this is an interesting idea and could potentially work with careful investigation, but it is rather outside of the scope of this paper. Many works considering layerwise methods (e.g. Huang et al. (2017)) resort  to a final end to end learning phases to make the results more competitive, but this portrays  the message that end-to-end learning is an absolutely necessary condition to get competitive performance. This is contrary to the message we are trying to convey: that greedy layerwise learning without end-to-end works much better than a large portion of the community would expect.

---

> ### Author Response · Authors · 2018-11-11
> **Response to R1 [2/2]**
>
> <<Maths in this paper is mostly decorative. >>
> We respectfully disagree.
> The two propositions are relevant, useful, and do not take up excessive space in the paper.
> The first proposition (though small) highlights a simple first question one would pose about such a system: “what are the conditions under which the loss would improve using layer-wise training?” The proposition shows that under a ReLU non-linearity and appropriate initialization the accuracy should improve with each new layer. We would like to highlight that this does not however answer the question of why the rate of improvement is fast in practice empirically, which is the surprising observation of the paper. We leave this open for further research.
>
> The second proposition highlights the fact that results from shallow networks can be extended to ones learned in a layerwise fashion. We demonstrate this explicitly for the case of provable optimality guarantees. Prop 3.2 highlights that if one has certain guarantees on the optimization for the sub-problem these can be directly extended to guarantees on the global problem. We re-emphasize that 1-hidden layer problems have been heavily studied recently in terms of provable optimization (Arora et al. (2018); Bengio et al. (2006); Bach (2014); Janzamin et al.(2015)
>  see discussions also in the penultimate paragraph of the Introduction and Sec 3.3.  Typically, extending those results from 1-hidden-layer networks to deep networks is extremely challenging, while in the layerwise learning case it is natural. We think this is not decorative, but an intuitive yet non-trivial result that explicitly highlights one of the major arguments of the paper: that theoretical results from shallow networks can be extended more easily.
>
> <<When comparing different models or training methods (e.g. layer-wise trained AlexNet and end-to-end trained AlexNet), it would make sense to do some hyperparameter search. It is very risky to conclude anything otherwise.>>
> In Sec 4.3, besides comparing to standard architectures we perform the most direct comparison possible of end-to-end learning on the same architecture with the same number of gradient updates on each parameter. The experiments in 4.1  do not compare directly the same architectures, but the aim of that Section was to show that the k=1 network is able to achieve AlexNet accuracies and exceed handcrafted feature performance. For completeness, we now have updated in the revision to show even in Sec 4.1 a direct end-to-end comparison for CIFAR-10 for the k=1 network with identical architecture, same amount of training epochs, same optimization hyperparameters per layer yields only a marginal improvement 89.7 vs 88.3.
>
> We do not believe based on the standards of reporting imagenet results that there is a risk we have somehow overfit imagenet. Few, if any, authors are able to extensively evaluate hyperparameters on this dataset with academic resources. We use identical optimization parameters commonly found in the literature  (batch size, SGD momentum and weight decay parameters) and in the examples of the pytorch framework for the auxiliary problem optimization. Our specific model and its auxiliary network have rather basic hyperparameters, not heavily tuned, which are further elucidated in the Appendix (which we have now extended as well). Sec 4.2 experiments use multiple initialization to assess variance.  A rather drastically new approach like this on imagenet is at a disadvantage if anything in terms of of hyperparameter optimization as we operate in a resource-limited academic setting and have been able to run only limited number of imagenet experiments (versus years of hyperparameter optimization done in the standard end to end learning setting by the community).
>
> We have updated the draft to emphasize these points.
>
> <<I would like to see a wall clock time comparison between this and end-to-end training.>>
>
> The basic layerwise method has the same number of gradient updates for each parameter as the end to end method (note that we use the same number of epochs for each layer training to ensure this) and theoretically has the same speed. Indeed this assumes the auxiliary network is of negligible compute with respect to the primary networks, which is the case for k=1 (but not the case in the current versions of k>1). However we have not focused on this aspect in our work  and believe future investigation aimed at speed can optimize the speed of the auxiliary network for k>1, for example by using fully connected layers instead of convolutions.

---

> ### Author Response · Authors · 2018-11-27
> **Additional Clarifications or Concerns?**
>
> Dear Reviewer,
> Thank you again for your review. Please let us know if we have addressed your concerns, especially those regarding novelty, and whether there is still any major outstanding point you would like clarified.

---

> > ### Comment · AnonReviewer1 · 2018-12-03
> > **after the rebuttal**
> >
> > I read the rebuttal, I thank the authors for answering my comments. Like I wrote before, I consider this paper correct and reasonable. However, I do think that these results are mildly novel, therefore I believe "marginally below acceptance threshold" is a well calibrated assessment in this case.

---

> > > ### Author Response · Authors · 2018-12-03
> > > **Clarification**
> > >
> > > Dear R1, We thank you for the response.
> > > We would like to ask if you can specify the insufficient novelty in the results that puts this under the publication bar for you? We believe to have addressed and disconfirmed all the claims of overlap to prior work. The results we show are not  present nor inferable from existing literature. Our paper and rebuttal argues extensively for their relevance. As you have not disagreed with any specific points raised regarding the relevance nor the major differences highlighted for each prior work mentioned, we kindly ask for a justification why the work should not be above the threshold for publication.

---

### Official Review · AnonReviewer2 · 2018-11-03
**Some interesting experiments and observations, but novelty is lacking**

**Rating:** 6
**Confidence:** 4

**Review:**



The authors propose to train deep convolutional neural networks in a layerwise fashion. This is contrary to the traditional joint end-to-end training of deep CNNs. As their motivation, the authors quote computational and memory benefits at the time of training in addition to being able to extend shallow-network analytical frameworks to the individual network layers thus allowing for a theoretical interpretation of their optima.

Their method is simple and clearly explained. (Note: In the 10th line on Pg. 4, is there a 'j' missing in the subscript of x^n?)
The experimental results are interesting. The authors are able to demonstrate 'some' architecture that, in solely a layerwise training, is able to show competitive results with respect to AlexNet when trained in an end-to-end manner on ImageNet. These results can seem questionable as both the architectures and training routines are being varied and hence the precise contribution of the layerwise training is unclear. However, as per my understanding, the aim of the paper wasn't to show that a layerwise training can work better that end-to-end training. The aim was, on the contrary to show, that 'even' a layerwise training can offer competitive performance for 'some' network and hence may come handy when memory is limited. Their underlying claim, which could be more clearly stated is that the memory benefits during training can be enjoyed when the individual layers of a network (net1) are smaller in parameter count as compared to another net (net2) although the net1 in totality maybe larger than net2. This is because net1 will be trained in a layerwise fashion, while net2 would be trained in an end-to-end manner. I would like to see the authors confirm or reject this understanding and rationalise their experimental regimen.

Further, I would like to know how their work compares to the following:
https://arxiv.org/abs/1703.07115
https://arxiv.org/abs/1611.02185

Finally, while the authors state that the layerwise training makes the individual layers amenable to theoretical analysis/interpretation, no such discussion is presented/initiated in the paper. The only analysis presented is on the ability of the individually trained layers to linear separate the data. To round the analysis, it should also be extended to the representations learned by end-to-end trained networks.

All in all, while the paper raises some interesting ideas, its execution in terms of a method that learns a classifier on each individual layer is rather simplistic. Don't get me wrong, simple can indeed be elegant, but at the minute the comparisons are not very convincing.

---

> ### Author Response · Authors · 2018-11-11
> **Response to R2 [1/2]**
>
> We thank you for the review  that will help us to improve the general quality of the paper. We answer below to each of your points.
>
> << As their motivation, the authors quote computational and memory benefits at the time of training  in addition to being able to extend shallow-network analytical frameworks to the individual network layers thus allowing for a theoretical interpretation of their optima.>>
> The computation and memory are indeed a potential benefit of greedy layerwise learning but it is not the only and not the primary motivation we have for greedy layerwise learning in the paper. Our paper emphasizes that it is easier to apply existing theoretical results to layerwise learning methods and that layerwise learning methods induce properties in the network that allow to more clearly study the behavior of each layer. The motivation of our work is to empirically determine if such greedy layerwise methods are viable (something not demonstrated in any related literature). In terms of the potential advantages of greedy layerwise methods we discuss extensively the ramifications for theoretical study of deep networks by giving a concrete example for optimization in Prop 3.2 and an empirical study that can only be done in this context in Sec 4.2. See also general comment and response to R1.
>
> <<Their method is simple and clearly explained. (Note: In the 10th line on Pg. 4, is there a 'j' missing in the subscript of x^n?) >>Thanks, we have fixed the typo
>
> << These results can seem questionable as both the architectures and training routines are being varied and hence the precise contribution of the layerwise training is unclear.>>
> We would like to emphasize that in Sec. 4.3, we obtain VGG-performance using an extension of the basic k=1 approach. We believe the reviewer is referring to the indirect comparison in Sec 4.1 to AlexNet without an e2e comparison. Please correct us if we have misinterpreted. We apologize for confusion for this and have addressed this point regarding the goal of 4.1 in the response we give to R1 and the general comment. To summarize briefly:
> -Sec 4.1 was aimed to show the most basic greedy layerwise method k=1 can already reach a performance barrier only obtained by deep learning methods on imagenet. For completeness we now report e2e comparison on CIFAR for same architecture.
> -Sec 4.3 already reports a very direct comparison, with the imagenet dataset, using end-to-end training and greedy layerwise training on the same architecture using the same number of gradient updates for each layer, and an architecture which performs as well as VGG in the e2e setting while being more compact.
>
> <<Their underlying claim, which could be more clearly stated is that the memory benefits during training can be enjoyed when the individual layers of a network (net1) are smaller in parameter count as compared to another net (net2) although the net1 in totality maybe larger than net2....>>
> This is indeed a stated potential practical advantage (and one we have benefited from in our own experiments). However, as emphasized in the paper in the introduction and Sec 3.3 and 4.3 there are many other reasons greedy layerwise learning is relevant. Our work is the only one that has shown this class of methods is potentially viable for real problems.
>
> <<Further, I would like to know how their work compares to the following:https://arxiv.org/abs/1703.07115 https://arxiv.org/abs/1611.02185>>
> Thanks for the references. We believe the contrast to these highlights well the contribution of our paper and have added them to the manuscript.
> The first one considers a layerwise supervised procedure and is indeed related in spirit. However, it shows numerical results only on the MNIST dataset of <96% and less than 50% on CIFAR. These are trivial accuracies even for those simple datasets, achievable without need for deep learning methods and do not in any way demonstrate that greedy layerwise construction can be used to build deep networks that show the kind of results that have made deep learning successful.
> The second reference is a layerwise scheme, but not greedy: although one layer is updated at a time the problem is solved jointly. This method is however an alternative to standard end-to-end back-propagation but we wish to bring attention that it is only shown to work as a strategy applied after typical end-to-end back-propagation. Our work is the first that shows greedy layerwise methods can achieve accuracies only obtained by e2e deep learning, and to the best of our knowledge it is the first alternative to end-to-end back-propagation that scales to imagenet and does not fall back on end to end learning (contrary to e.g. the 2nd reference which does).

---

> ### Author Response · Authors · 2018-11-11
> **Response to R2 [2/2]**
>
> <<Finally, while the authors state that the layerwise training makes the individual layers amenable to theoretical analysis/interpretation, no such discussion is presented/initiated in the paper.>>
> A substantial portion of the paper is devoted to emphasizing this aspect. First, as discussed in the Introduction and Sec. 3.3, 1-hidden layer networks are widely studied and have a large number of results that theoreticians struggle to extend to deeper networks under normal training settings (see extensive reference). The mathematical analysis, particularly Prop 3.2, is aimed at explicitly demonstrating how results of 1-hidden layer, or more generally sub-problems, can be applied to the cascade when it’s constructed greedily, showing that optimization results can be directly extended to deep networks from shallow ones. Please also see the response to R1.  Secondly, we also discuss how layerwise objective specification can allow to test if a specific layer-level property can be a sufficient condition to account for high accuracy when layers with this property are cascaded. We devote a substantial amount of empirical analysis to study this for the case of linear separability in Sec 4.2.
>
> We also point out two related submissions to this same conference: https://openreview.net/pdf?id=H1GLm2R9Km , which considers a greedy layerwise objective and focuses on the theoretical aspects. Another paper: https://openreview.net/pdf?id=r1Nb5i05tX , attempts to use layerwise objective specification to determine the effectiveness of the information bottleneck principle (similarly to our section 4.2 for linear separability).  These, as the works we cite, rely on experimental settings and accuracies not of high interest to the deep learning community. However, it shows that there is interest in greedy layerwise learning beyond our work in these theoretical and interpretation contexts.
> Yet, the big question up until now is whether such an approach can work on real problems that are of interest to the community. Our paper answers this question for the first time: there is no previous result, to the best of our knowledge, to indicate that the community should expect these methods to be more than theoretical ideas that won’t work in practice.
>
> <<The only analysis presented is on the ability of the individually trained layers to linear separate the data. To round the analysis, it should also be extended to the representations learned by end-to-end trained networks. >>
>
> Multiple works have shown linear separability results for end to end trained network (Zeiler & Fergus, 2014; Oyallon, 2017, Jacobsen et al., 2018). As emphasized in the paper this analysis from previous works is incomplete as the linear separability might be an indirect consequence of secondary objectives implemented by the layers. Direct layerwise objective specification permits to decouple empirically observed properties of layers to allow claims that they are sufficient to achieve a certain objective. We believe other proposed layerwise mechanisms e.g. the recently proposed Information Bottleneck can be studied this way.
>
> However, we do agree with the reviewer that it would be good to compare the end-to-end linear separability directly on the same architecture! We have added this comparison by adding a direct comparison in Figure 2 to the end-to-end network in the new revision. It illustrates the progressive linear separability properties are similar for end-to-end networks to k>1 training. It thus seems to be a generic idea.
>
> <<All in all, while the paper raises some interesting ideas, its execution in terms of a method that learns a classifier on each individual layer is rather simplistic. Don't get me wrong, simple can indeed be elegant, but at the minute the comparisons are not very convincing.>>
> This work has shown previously unexpected results with greedy layerwise learning that are not directly suggested by any others. The simplicity has been an important point that has allowed us to carry out this proof of concept and we were surprised nothing more complex was needed. We certainly think this shouldn’t be counted against the message of the work, which is not focused on the specific method but on whether greedy layerwise strategies can achieve high performance. Please see our general comment for a more in depth discussion.

---

> ### Author Response · Authors · 2018-11-27
> **Additional Clarifications or Concerns?**
>
> Dear Reviewer,
> Thank you again for your review. Please let us know if we have addressed your concerns, especially those regarding novelty, and whether there is still any major outstanding point you would like clarified.

---

### Public Comment · (anonymous) · 2018-10-10
**Facinating result!**

The paper shows that greedy layerwise learning of neural networks can achieve state of the art results on large scale perceptual datasets such as imagenet and cifar10. This is a exciting result for (at least) three reasons:

1. Greedy training has huge benefits in terms of training time and memory
2. As in each greedy step a depth-two network is trained.  The algorithmic problems that underlie the training process become far easier and better understood.
3. The fact that the greedy approach achieves state of the art result establishes a great insight, as it challenges the so-common end-to-end approach.

---

### Public Comment · (anonymous) · 2018-10-11
**Shallower, yet (much) wider**

It is well known that wider networks have greater expressiveness and better performance. Therefore it comes to me as no surprise that the networks (with its retro training techniques) proposed in this paper matches the performance of their deeper baselines. As a illustration, here is a summary of the size of the network parameters used with the CIFAR-10 dataset:

(convolution layer size: out_channels * in_channels * kernel_w * kernel_h; fully-connected layer size: out_features * in_features; biases not counted)
Author_CIFAR: 256*3*3*3 --> 512*256*3*3 --> 512*512*3*3 --> 1024*512*3*3 --> 1024*1024*3*3 --> 10*4096 (total weight parameters=17.7M, CIFAR-10 acc=88.3%)
AlexNet_CIFAR: 64*3*5*5 --> 64*64*5*5 --> 64*64*3*3 --> 64*64*3*3 --> 10*3136 (total weight parameters=0.2M, CIFAR-10 acc=89% w/ local response norm, 87% w/o local response norm [1])

Basically, the proposed method achieves comparable performance with ~100x more parameters than AlexNet.

Network this large can easily achieve >95% acc on CIFAR-10 using some recent architectures (see, e.g. [2] where they reported 96.4% acc with 15.3M parameters). Yet I am more interested to know the results of the following two simple comparisons:
1) test accuracy of the proposed network architecture with end-to-end training;
2) test accuracy of AlexNet trained with the method proposed in this work.
They would comprise enough empirical evidence for me to decide the value of this work.

[1] https://code.google.com/archive/p/cuda-convnet/
[2] https://github.com/liuzhuang13/DenseNet#results-on-cifar

---

> ### Author Response · Authors · 2018-10-14
> **Calculation for Alexnet mistaken, we do compare to end to end nothing obvious about our results**
>
> We thank you for your interest in our work and hope we can address your concerns.
>
> By our count, Krizhevsky's Alexnet (https://code.google.com/archive/p/cuda-convnet/) uses 3.7M parameters, which break down into 64*3*5*5+64*64*5*5+64*64*3*3*8*8+64*32*3*3*8*8+32*8*8*10 (including the locally connected layers), which is around 1/4 of ours, not 1/100. Similarly, our ImageNet model for k=1 has roughly 4x the parameter count of AlexNet for ImageNet. A factor 100 would be impossible (at least with any typical resources) nor is it sufficient even in that case to explain the observed phenomenon as we will clarify below. On the other hand for our models k=3, M_f=512 the size is quite smaller (1/3)  that of VGG-11, and we directly compare this model in end to end training in Sec 4.3 among other comparisons to end to end.
>
> Our paper does not only aim to propose a specific method but also to ask an empirical question about the limits of layerwise objectives. Thus 4.1 and 4.2 use a very restrictive but interesting setting (k=1) that focuses on this question. Section 4.3 deals with practical applications of our results (such as comparing to  end-to-end training).  Thus the comparisons to end-to-end models and parameter efficiency were limited to Section 4.3. The last two major sub-paragraphs of 4.3 explicitly discuss these and include end-to-end trained imagenet and cifar models compared to layerwise training.
>
> The model assessed in Sec 4.1 was constructed for the aim to understand the limits of 1-hidden layer models. As such they are not necessarily the best for end to end training comparisons. For example they do not use batchnorm (unnecessary in k=1) and they do exploit the fact that as we only backprop through the top layer being processed, which makes it possible to fit larger models onto the same hardware. For completeness, we now evaluated end-to-end training for CIFAR-10 k=1 (with CIFAR settings described in Sec 3) and it achieves 89.7 (recall there is no batchnorm), and will include this in a revision. Indeed it is a bit better than the k=1 based model (though not the 95% you claim), but we do not believe it affects any arguments in Sec 4.1. Sec 4.2 shows that a big improvement can be obtained (for the same sized networks) when using k>1.  An open question left by the work is if the k=1 model can apply a better objective that permits them to match the performance of k>1 models, which are shown to be competitive with end to end in Sec 4.3
>
> While we agree with you that wider networks can often lead to better performances, the scale of those improvements is not remotely sufficient to account for our results in Sec 4.1. Take the example in Wide Residual Networks (Zagoruyko et al), for Resnet-18 increasing the model size 4 fold (layer width 2x) moves the top5 accuracy from 89.07 to 91.94.   On the other hand the 1-hidden layer CNN model we start with, for imagenet alone obtains only 23% top 5 accuracy (observe this  in Fig 3 of the Appendix). Simply making the cnn layer wider here without using a cascade is unlikely to improve to 80% top 5, nor is it practical to even attempt to increase this model by 100 or 1000 times with finite resources. On the other hand from Fig 3 of Appendix observe there is a rapid increase  (near linearly in the first layers) to reach an accuracy of 80% top 5.  This rapid rate of progressive improvement of the cascade is not expected from any previous result we are aware of and we think there is nothing obvious here.

---

### Public Comment · (anonymous) · 2018-10-12
**Similar to Deep Growing Learning**

The paper shows  competitive results to end-to-end training of CNNs. However,  a related work called Deep Growing Learning [1] shares a similar idea with this paper. The same points are:

1) layer-wise training
2) keeping all other parameters fixed when training j-th layer.

The  only different point is that deep growing learning  focuses on semi-supervised learning while this paper focuses on supervised learning. Deep growing learning only shows empirical evidence  for supervised learning on their experiments.

[1] http://openaccess.thecvf.com/content_ICCV_2017/papers/Wang_Deep_Growing_Learning_ICCV_2017_paper.pdf

---

> ### Author Response · Authors · 2018-10-14
> **Thank you for the reference**
>
> Dear Anonymous commenter,
>
> Thank you for bringing this paper to our attention, we will add it to our references in future revisions. The paper you reference clearly indicates in multiple places including the Algorithm block line 17 that after each layer is added a global fine-tuning is performed. This is very distinct from both our method and the high level questions we try to answer in our paper. We believe that it falls directly into the category of papers already discussed in the last paragraph of our Related Work section. The paper also by nature focuses on small scale datasets (small sample MNIST and CIFAR) and there is also some relation to the structure adapting methods discussed in our related works, which we think is a great application of layerwise learning but is not the topic of focus of our paper.  We do not believe there is an overlap to the contributions listed in the end of the Introduction in our paper.
>
> We thank you again for your comment and bringing this reference to our attention.

---

### Author Response · Authors · 2018-11-11
**Significant novelty lies in empirical results, not specific method. Nothing obvious or covered in literature**

We thank the reviewers for their reviews and the time spent on the manuscript. We have have used them to further improve the manuscript. We give a general answer addressing a point that was raised in more than one review:

R1 and R2 have indicated that this work is of potentially limited novelty. We believe that the novelty of the scientific results and their potential contribution to the literature  are being confounded with the specific method we used, which turned out not needing to be complex. In the past, greedy layerwise methods have been attempted, with limited success. We have successfully revisited this idea, demonstrating a proof of concept which challenges the prevalent assumption that joint learning is essential. E.g. in  [1] it is claimed: “One of the most interesting challenges raised by deep architectures is: how should we jointly train all the levels?”.  On the other hand we ask a more underlying question: “Is jointly training all the levels essential?”.  We do not believe there is any remotely similar empirical or theoretical result that can suggest that layerwise training with even 1-hidden-layer problems can work on tasks such as imagenet, and that there is nothing obvious/expected about our results. In short, we strongly defend the novelty and relevance of this work.

We recall a few key contexts and then re-iterate our contributions.
In machine learning it is common that methods are completely ignored both by theoreticians and practitioners until they are shown to be competitive on real problems.  Indeed many of the famous break-through works in the recent deep learning literature consider ideas that have been previously examined in related forms but not clearly shown to work well.  An example is the AlexNet itself which can be seen as a trivial extension of the LeNet and other CNNs from the 90s. The contribution was to show their utility on a large-scale problem. Another example is LSTM/RNNs, an old idea long believed to be impractical, that in the early 2010s was shown to work using largely similar ideas as older works with minor modifications.  Supervised layerwise learning methods have indeed been proposed in some forms as discussed extensively in our work and a method cited by the reviewers. Yet, to the best of our knowledge on standard tasks such as image classification there is not a single instance of such a method that works better than even handcrafted image descriptors without resorting to end-to-end learning (82% on CIFAR is the best result we could find). No greedy layerwise method has been shown to work in any capacity on imagenet at all to the best of our knowledge.

In this context we reiterate our contributions emphasizing the latter two:
- To the best of our knowledge we propose the first greedy layerwise strategy for CNNs, that can scale (computationally) to large images.
- We show that the most barebones layerwise-only method, of training 1-hidden layer at a time, can well exceed the performance of any alternative method on imagenet, and the famed AlexNet. AlexNet remains the minimal deep learning accuracy which exceeds handcrafted descriptor performance, and by exceeding it we show for the first time that a simple layerwise method can attain the famed accuracy boost of deep learning models over non-deep learning models.
- We show that extensions of this strategy can become competitive with end-to-end learning even at much higher accuracies such as those of the VGG family

We have updated the introductory section to further emphasize the empirical demonstration over the precise method as the contribution.  As requested by R3 we add an anonymized (preliminary) repository which can be used to reproduce key experiments in the paper: https://anonymous.4open.science/repository/75115ffe-d110-4814-ade0-060dd12b9f7e/

[1] Representation Learning: A Review and New Perspectives, Yoshua Bengio, Aaron Courville, Pascal Vincent, 2014

---

### Public Comment · ~Enrique_Salvador_Marquez2 · 2018-11-19
**Similarities with Cascade Learning publication**

Very interesting work.

Can I point to some similar work we published recently, on a supervised layer-wise training algorithm, also motivated by Scott Fahlman's Cascade Correlation idea. We refer to our approach as Deep Cascade Learning [1]. You might find this relevant.

Comparing these, I note:

1) Both are greedy layer-wise, and on every iteration it trains one convolutional layer.
2) Cascade Learning does not use a dimensionality reduction block, while in this work it uses an extra convolutional layer and pooling operators. However, in our most recent Cascade Learning implementations we have found the need of using such techniques to reduce the number of parameters in the softmax layer.
3) To our knowledge, the ensemble of classifiers at multiple stages of the network is novel, and it is an interesting idea to explore.
4) In our work we show other advantages of such layer-wise training of deep networks. Specifically, we show complexity advantages due to a reduction in the forward passes of already trained layers, and lower convergence variance in comparison with end to end training.
5) We also perform an empirical gradient analysis to observe how approximating the output layer to every stage of the network may tackle the vanishing gradient problem.

In recent (unpublished work), we also find that this is a nice approach for transfer learning and it drastically reduces the hardware requirements to tune the feature extractor.

[1] Marquez, Enrique S., Jonathon S. Hare, and Mahesan Niranjan. "Deep cascade learning." IEEE Transactions on Neural Networks and Learning Systems (2018).

---

> ### Author Response · Authors · 2018-11-19
> **Thanks for the reference**
>
> Thanks for your comment Enrique and pointing out this concurrent work. We’ll add your recent paper to our references. We point out several major distinctions:
>
> 1) One setting covered and highly emphasized in our work considers the case where the auxiliary network is simply a linear operator, making the auxiliary problem a 1-hidden layer network, which is not covered in your work. It is a restrictive and challenging setting for us to consider but we argue in the paper  that this kind of construction has interesting theoretical properties, due to the unique nature of the shallow 1-hidden layer problem. The fact that greedy learning is shown in our paper to work in this setting leaves multiple open questions for the community. The models used by your work is more similar to the ones we study under the setting of k>1 (in Sec 4.3).
> 2)  A major distinction of our work to previous ones on greedy layerwise learning is that we focus on accuracies and settings that have only been achieved by supervised deep learning methods -  the settings that have made deep learning so important today. We show that it is possible to do greedy layerwise learning in these regimes without any defaults to end to end learning. Achieving these high performances is necessary for a meaningful evaluation whether end-to-end learning is a necessary, required procedure for training deep networks. Only then can one potentially convince the community of the usefulness of greedy layerwise techniques. For example, even on the CIFAR-10 dataset, handcrafted image descriptors and unsupervised learning can already achieve accuracies of 82/84% (Oyallon, et al 2017). On imagenet there is not a single instance to our knowledge of greedy methods being applied, much less achieving accuracies associated with deep learning models like VGG. We believe our work is the first to consider these regimes (e.g. 91%+ on CIFAR, 88+ top 5 imagenet).
> 3) As you noted, from a technical point of view the architecture in your work that flattens the convolution before applying FC layers would be challenging to scale to large images. The averaging operation we introduce for greedy training of CNNs is quite important to permit scaling to imagenet.
>
> Regarding the idea to train fewer iterations earlier layers, we have also experimented with this at early stages and agree that this is an interesting direction! For more direct comparisons in our paper we stuck to the more straightofrward use of same training strategy for all layers and the end to end comparison
>
> We thank you very much for your comment, we are excited to see that the community still has interest in these methods! Scaling on ImageNet those results, as you can guess, is  rather challenging and we hope can benefit this direction of research.

---

### Author Response · Authors · 2018-11-19
**Revision Update**

We thank again the reviewers for their comments and review of the paper. It has helped us to revise the manuscript. We itemize the revisions in our manuscript with the following (primary) changes based on the reviewer input:
- Revised first paragraph of introduction to emphasize the high level question approached by our empirical work
- Added references from  comments/reviews
- A comparison to end-to-end learning for the k=1 model in Sec 4.1
- A direct comparison to e2e networks in the linear separability studies of 4.2 (Fig 2)
- Ablation studies for downsampling operation in the Appendix
- Ablation studies for width and details of parameter counts in Appendix
- Transfer learning on Caltech 101 in Appendix

---

### Public Comment · (anonymous) · 2018-12-09
**Seems to be similar to "Forward Thinking" submission from NIPS 2017**

How does this method compare to "Forward Thinking: Building and Training Neural Networks One Layer at a Time" (https://arxiv.org/pdf/1706.02480.pdf)? Although the experiments here are much more thorough, the underlying ideas seems similar. A comparison (and definitely a reference) would be nice.

---

> ### Author Response · Authors · 2018-12-09
> **Similar related work covered in our paper, MNIST is solvable with single hidden layer**
>
> Thank you for the reference! Concerning core contributions, there is little to no overlap with our work and this (unpublished) arxiv paper (which is not a NeurIPS paper as implied).  This paper fits clearly in the category of work described in 3rd paragraph of our related work section, and we do not believe it adds much beyond those works to the central question in our paper. Unlike our work, it  does not demonstrate  the viability of greedy methods on problems where deep networks are needed. The experiments in the reference are on MNIST, which is solvable with a single hidden layer (e.g. https://arxiv.org/abs/1412.7259v1 ) or with a host of hand crafted feature methods, whereas Imagenet is inconceivable to analyze with a single hidden layer - reasonable performance requires a deep cascade (see e.g. our Figure 3).  Note also we have an extremely extensive literature review in our paper (to be extended even further based on OpenReview comments) and we very unambigously establish in the paper that we do not introduce the high level idea of greedy learning of NNs

---

### Meta-Review · Area_Chair1 · 2018-12-14
**not ready for acceptance**

**Confidence:** 5
**Recommendation:** Reject

**Metareview:**

The paper discusses layer-wise training of deep networks. The authors show that it's possible to achieve reasonable performance by training deep nets layer by layer, as opposed to now widely adopted end-to-end training. While such a training procedure is not novel, the authors argue that this is an interesting result, considering that such a training procedure is often dismissed as sub-optimal and leading to inferior results. However, the results show exactly that, as the performance of the models is significantly worse than the state of the art, and it is unclear what other advantages such a training scheme can offer. The authors mention that layer-wise training could be useful for theoretical understanding of deep nets, but they don’t really perform such analysis in this submission, and it’s also unclear whether conclusions of such analysis would extend to deep nets trained end-to-end.

In its current form, the paper is not ready for acceptance. I encourage the authors to make a more clear case for the method: either by improving results to match end-to-end training, or by actually demonstrating that layer-wise training has certain advantages over end-to-end learning.

---

> ### Author Response · Authors · 2019-01-02
> **Some Remarks on the Meta-Review**
>
> We thank you very much for your meta review as well as the reviewers’ work that will help us to revise the paper. However, we disagree with several statements and conclusions in this meta-review, some of which we believe do not take into account our rebuttal and substantial portions of the paper. As this meta-review also brings multiple new points of criticism that are not made by the reviewers we would like to conclude by addressing the major ones.
>
> “the performance of the models is significantly worse than the state of the art”. We show for the CIFAR-10 dataset an identical performance to end-to-end training for the same network architecture (Sec 4.3  last par.). We show for imagenet performance of 88.5 compared to a baseline of 90.1 (Sec 4.3;Tab 1, on par with VGG-nets). These results  are more than sufficient to support our claims. There are many instances of alternative training procedures  that can be made to work mildly on small datasets like MNIST or even CIFAR (obtaining 80% accuracy), but simply do not work on large datasets like Imagenet (Sec 4.1 par 1),  sometimes obtaining accuracies below 20% top 5.  Gaps of 50%+ to the state of the art are clearly  not the same as a 1% difference on imagenet, which  we demonstrate  has many ways to be closed.  We have also discussed repeatedly in our paper that the imagenet accuracies obtained by our method correspond to those of models that have been used for countless downstream applications and even deployed in industrial settings. To the best of our knowledge and in communications with members of the community obtaining VGG like accuracies with  these methods is  not  expected. The review of R3 appears to agree with this, even asking for code during the review process to verify the claims.
>
> “such a training procedure is not novel”. We have addressed this statement in our rebuttal, pointing out that even the specific instantiation of our method adds simple but novel aspects that turn out essential for scaling such methods to large images (and datasets).
>
> “could be useful for theoretical understanding of deep nets, but they don’t really perform such analysis in this submission” This is inaccurate. Prop 3.2 is simple approach to fill the gap between 1-hidden layer training literature and deep models with practically relevant performance, by claiming the tractability of the optimum for the deep network optimization, showing how one can use our models to directly connect to existing theoretical results. There are multiple works that conclude that it is extremely hard to show similar optimization results for standard end-to-end learned deep networks. We also point out our empirical study of linear separability (Sec 4.2) that makes several novel observations about the role of this phenomenon that require layerwise training to observe (Sec 4.2, par. 5). We are not aware of any claims by the reviewers or meta-review that show or argue that these contributions are not novel nor relevant.
>
> “unclear whether conclusions of such analysis would extend to deep nets trained end-to-end.”
> (a)If a model and training procedure is easier to analyze and is shown to produce a highly useful model (good enough  to use in industrial settings or downstream applications in countless vision papers)  it is then certainly preferable to analyze  over an end to end approach. We have multiple published references in our paper where theoretical work do just this: analyze greedily trained models because they simplify the analysis. Knowing greedily trained models can produce useful models is thus a desirable fact to demonstrate. (b) In our paper and rebuttal we discuss in detail that hypotheses for layerwise functionality (e.g.  progressive linear separability or mutual information) in end to end trained networks can be more directly evaluated via layerwise training. Please see our Introduction and Sec 4.2 for more detailed discussion of this point.
>
> “it is unclear what other advantages such a training scheme can offer”: Reviewers observed several advantages of the method (R1,bulletpoint strength ; R3 second comment ; R2 par. 5-7), that we also mention (Sec 3.3; Sec 2; Sec 4.3 ; Sec 5). Furthermore, (Sec 2; end of Sec 4.3) is devoted to motivating specific advantages from multiple angles; no precise argument is made against any of those points in this review process. (Please also see our answer to R2(par 4, 9,11) and R1(par 3, 5) )